# Anti-Allergic Diarrhea Effect of Diosgenin Occurs via Improving Gut Dysbiosis in a Murine Model of Food Allergy

**DOI:** 10.3390/molecules26092471

**Published:** 2021-04-23

**Authors:** Chung-Hsiung Huang, Chorng-Liang Pan, Guo-Jane Tsai, Chun-Ju Chang, Wei-Chung Tsai, Shueh-Yu Lu

**Affiliations:** 1Department of Food Science, National Taiwan Ocean University, Keelung 20224, Taiwan; b0037@mail.ntou.edu.tw (C.-L.P.); b0090@mail.ntou.edu.tw (G.-J.T.); chunju@mail.ntou.edu.tw (C.-J.C.); 10932036@mail.ntou.edu.tw (W.-C.T.); 0043a029@email.ntou.edu.tw (S.-Y.L.); 2Center of Excellence for the Oceans, National Taiwan Ocean University, Keelung 20224, Taiwan

**Keywords:** allergic diarrhea, cytokine/chemokine network, diosgenin, dysbiosis, gut microbiota, food allergy, T-cell response

## Abstract

Although the anti-allergic and prebiotic activities of diosgenin have been reported, the influence of diosgenin on intestinal immune and epithelial cells remains unclear. As the gut microbiota plays an important role in allergic disorders, this study aimed to investigate whether the anti-allergic diarrhea effect of diosgenin occurs via improving gut dysbiosis. In a murine food allergy model, the density of fecal bacterial growth on de Man, Rogossa and Sharpe (MRS) plates was diminished, and growth on reinforced clostridial medium (RCM) and lysogeny broth (LB) agar plates was elevated. However, the oral administration of diosgenin reduced the density of fecal bacteria and ameliorated diarrhea severity. Concordantly, reshaped diversity and an abundance of fecal microbes were observed in some of the diosgenin-treated mice, which showed a milder severity of diarrhea. The relevant fecal strains from the diosgenin-treated mice were defined and cultured with Caco-2 cells and allergen-primed mesenteric lymph node (MLN) cells. These strains exhibited protective effects against the cytokine/chemokine network and allergen-induced T-cell responses to varying degrees. By contrast, diosgenin limitedly regulated cytokine production and even reduced cell viability. Taken together, these findings show that diosgenin per se could not directly modulate the functionality of intestinal epithelial cells and immune cells, and its anti-allergic effect is most likely exerted via improving gut dysbiosis.

## 1. Introduction

Food allergies are characterized by abnormal immune reactions to ingested food allergens [1]. The intestinal tract is the main site affected by food allergens, and diarrhea is the most common clinical sign. The prevalence of food allergies has increased worldwide in recent years, resulting in a considerable public health and economic burden [1]. The current management of food allergies mainly relies on the avoidance of allergens. Therefore, researchers have attempted to develop effective measures to prevent or treat food allergies. Although the key feature of a food allergy is a T-helper 2 (Th2)-dominant, allergen-specific immune response, a failure to establish—or a breakdown in the maintenance of—oral tolerance may be responsible [1]. Oral tolerance can be influenced by many factors. Among these factors, gut dysbiosis is of great significance [1]. Clinical epidemiologic studies have pointed out that decreased microbial pressure is correlated with the increased prevalence of allergies in Western countries [2]. In accordance with epidemiologic results, experimental evidence substantiates the notion that a disease-associated microbiota is crucial for the pathogenesis of food allergies [3]. In addition to dysbiosis, homeostasis of the cytokine/chemokine network in the intestinal microenvironment is also a major regulatory mechanism against the development of food allergies [4]. In general, a food allergy is determined by an interaction between genetic background and environmental influences. Many studies have found that environmental factor-induced allergic or anti-allergic effects are mediated by epigenetic mechanisms [5,6,7,8,9]. For example, the decreased methylation of genes encoding Th2 cytokines and increased methylation of the gene encoding Th1 and regulatory T-cell (Treg) cytokines were observed in individuals with a cow’s milk allergy [10]. In a whey protein/cholera toxin-induced murine model of food allergy, the elicited allergic response was correlated with a lower population of Treg and Th17 cells, accompanied by the downregulation of H3 and H4 histone acetylation at Treg and Th17 loci [11]. Moreover, Suzanne et al. have assessed histone acetylation at the promoter regions of Th1-, Th2-, Th17-, and Treg-related genes in splenocyte-derived CD4+ T cells and mesenteric lymph nodes (MLNs) to determine the role of epigenetic mechanisms in the allergy-protective effect against raw milk. After raw milk exposure, the histone acetylation of Th1-, Th2-, and Treg-related genes in splenocyte-derived CD4+ T cells was upregulated. After allergy induction, the histone acetylation of Th2 genes was reduced [12]. In addition to systemic splenocyte-derived T-cells, histone acetylation in the local MLN was also addressed, and similar effects were observed [12].

Diosgenin, a major yam-derived steroidal sapogenin, is one of the active ingredients contributing to the biological effects of the Chinese yam. It has been reported that the oral administration of diosgenin to rodents could alleviate intestinal inflammation and allergic reactions, providing a line of scientific evidence to support the beneficial effects of the Chinese yam on the immune and gastrointestinal systems [13,14]. Although the oral administration of diosgenin could attenuate allergic responses in mice with an ovalbumin (OVA)-induced food allergy, not all allergic mice treated with diosgenin could be protected from allergic diarrhea. Additionally, it has been reported that diosgenin could augment the growth of fecal *Lactobacillus reuteri* and *L. murinus*, and these two strains were able to attenuate food allergies [15,16,17]. Dong et al. also substantiated the notion that the modulatory effect of diosgenin on the intestinal microbiota could facilitate antitumor immunity and the therapeutic efficacy of the PD-1 antibody [18]. It should be noted that the oral bioavailability of diosgenin is very low [19], and information pertaining to the direct impact of diosgenin on intestinal epithelial and immune cells is not available yet. Thus, we hypothesized that the beneficial effects of diosgenin on the immune and gastrointestinal systems are potentially exerted via its action on the gut microbiota.

Recently, the role of the gut microbiota in health and disease has been extensively studied using culture-independent molecular methods [4,20]. Due to its many advantages, next-generation sequencing (NGS) of the 16S rRNA gene is now the most widely used tool to investigate the gut microbiota. Compared to traditional culture methods, NGS, a high-throughput sequencing tool, is more sensitive and can provide a large sampling depth for the detection of low-abundance taxa. Thus, the results of NGS analysis not only deepen our understanding of the gut microbiota in the progression of allergic diarrhea, but also elucidate whether bioactive food ingredients are able to improve dysbiosis. In an OVA-induced murine model of food allergy, the fecal microbiota displayed a significantly different profile between normal and allergic mice, as observed by NGS analysis. The relevant fecal microbes involved in allergen-induced dysbiosis promoted the intestinal cytokine/chemokine network and allergen-specific T-cell immune responses [4]. On the other hand, the anti-allergic effect of diosgenin was also substantiated in the same murine model, although diosgenin treatment (200 mg/kg b.w. daily) only partially attenuated allergic diarrhea [13]. Therefore, the same murine model was employed to investigate whether the anti-allergic diarrhea effect of diosgenin is mediated by improving gut dysbiosis. Firstly, the differences in fecal microbiota among allergic mice treated with or without diosgenin were explored using traditional plating methods and NGS technology. Secondly, relevant fecal microbes in diosgenin-treated mice were defined, and the impact of these fecal strains on the intestinal cytokine/chemokine network and allergen-specific T-cell responses was evaluated in Caco-2 cells and OVA-primed MLN cells. Finally, the results obtained from cells cultured with fecal strains were compared to those obtained with diosgenin.

## 2. Materials and Methods

### 2.1. Chemicals, Reagents and Cell Lines

All the chemicals, including diosgenin, and the reagents were purchased from Sigma Chemical (St. Louis, MO, USA) unless otherwise stated. The reagents for cell culture and bacterial culture were purchased from GE Healthcare Life Sciences (Marlborough, MA, USA) and BD Diagnostics (Sparks Glencoe, MD, USA), respectively. The ELISA reagents and standard were purchased from eBioscience, Inc. (San Diego, CA, USA), while 10% neutral buffered formalin was purchased from Leica Biosystems Richmond Inc. (Richmond, IL, USA). The Caco-2 cell line (ATCC^®^ HTB-37™) was maintained in DMEM (L-glutamine, high glucose; Corning Inc., New York, NY, USA) supplemented with 10% fetal bovine serum.

### 2.2. Mice and Ethics Statement

For this study, 4–5-week-old female BALB/c mice were obtained from the National Laboratory Animal Center in Taiwan. All mice were housed at the Laboratory Animal Facility of the National Taiwan Ocean University (NTOU). On arrival, mice were randomly transferred to plastic cages containing sawdust bedding and quarantined for 1 week. Mice were housed in a temperature- (23 ± 2 °C), humidity- (50 ± 20%) and light- (12 h light/dark cycle) controlled environment. Except on the days of allergen challenge (described below), food and water were supplied ad libitum. The study was conducted in accordance with the Declaration of Helsinki, and the protocol was approved by the National Taiwan Ocean University Institutional Animal Care and Use Committee (NTOU-IACUC-107026).

### 2.3. Protocol of Animal Experiments

The employed experimental protocol and dosing regimen for diosgenin and OVA sensitization and challenge have been reported previously [13,15]. Briefly, the mice (8 mice per group) were divided randomly into the following groups (Figure 1): naive (NA), OVA-sensitized and challenged (OVA) and diosgenin-treated and OVA-sensitized and challenged (OVADIO). Diosgenin (200 mg/kg b.w.) was suspended in olive oil as a fine suspension and administered daily to mice by oral gavage throughout the experiment. Except for mice in the NA group, each mouse was sensitized with OVA by intraperitoneal injection using 0.1 mL of sensitization solution containing 50 μg of OVA and 1 mg of aluminum potassium on day 3 and later boosted with a double dose on day 17, followed by repeated challenge with OVA (50 mg) by gavage every other day from day 31 to day 41. Mice were deprived of food for 3 h before the administration of diosgenin on the days of challenge and were challenged 1 h after the administration of diosgenin. Allergic diarrhea was apparent 30–60 minutes following the challenge, and it was assessed by a severity score of the fecal form from 0 to 3: 0, no fecal matter or solid state; 1, funicular form; 2, slurry; 3, watery state [21]. The mean score indicates the average severity score from two mice in the same group with almost the same severity of diarrhea throughout the experiment. The individual scores are shown in Appendix A. The mice were euthanized 3 h after the last OVA challenge. Subsequently, fresh fecal samples from the two mice were harvested and pooled together for NGS analysis and bacterial culture. Therefore, there were 4 fecal samples per group, and the samples were labeled as 1, 2, 3 and 4 (i.e., the fecal samples of the OVADIO group were named OVADIO1, OVADIO2, OVADIO3 and OVADIO4). Concurrently, the duodenal tissue was isolated for the preparation of hematoxylin and eosin (H&E)-stained sections.

### 2.4. Next-Generation Sequencing (NGS) Analysis

Fecal DNA extraction was performed using the QIAamp DNA Stool Mini Kit (QIAGEN, Germantown, MD, USA) following the manufacturer’s recommendations. Metagenomic studies were performed by analyzing the prokaryotic 16S rRNA gene. In this study, 2.5 μL of DNA was used to set up the first PCR with 0.2 μM V3 + V4 forward and reverse primers (forward: TCGTCGGCAGCGTCAGATGTGTATAAGAGACAGCCTACGGGNGGCWGCAG, reverse: GTCTCGTGGGCTCGGAGATGTGTATAAGAGACAGGACTACHVGGGTATCTAATCC) and 12.5 μL 2× Kapa HiFi HotStart ReadyMix (KapaBiosystems, Wilmington, MA, USA) in 25 μL reactions. The PCR cycling conditions were 3 min at 95 °C, 25 cycles of 30 s at 95 °C, 30 s at 55 °C and 30 s at 72 °C, followed by 5 min at 72 °C. The amplified DNA was purified with Agencourt AMPure XP Reagent beads (Beckman Coulter Inc., Brea, CA, USA). The second PCR was set up to add indexes to the amplified DNA by adding 5 μL of purified DNA to 25 μL 2× Kapa HiFi HotStart ReadyMix (KapaBiosystems), 5 μL Nextera XT Index 1 and 2 primers (Illumina, San Diego, CA, USA) in 50 μL reactions. The second PCR reaction was set at 3 min at 95 °C, 8 cycles of 30 s at 95 °C, 30 s at 55 °C and 30 s at 72 °C, followed by 5 min at 72 °C, followed by another stage of Agencourt AMPure XP Reagent bead purification (Beckman Coulter, Inc., CA, USA). We used qPCR (Kapa SYBR FAST qPCR Master Mix) to quantify each library using the Roche LightCycler 480 system and pooled them equally to 4 nM for the Illumina MiSeq NGS system (Illumina, CA, USA). More than 100,000 reads with paired-end sequencing (2 × 300 bp) were generated. The sequence data were then processed using the QIIME2 package version 2019.4 [22]. Forward and backward reads were joined, and the tool was used to select closed-reference OTUs from the GreenGenes database (May 2013 version). OTU representative sequences were selected afterwards with a 97% similarity threshold.

### 2.5. Culture of Fecal Bacteria

A previously described plating method was employed, with minor modifications [23]. Briefly, after six rounds of OVA challenge, the fresh fecal samples collected from mice were weighed, suspended and serially diluted with buffered peptone water and cultured in triplicate on de Man, Rogossa and Sharpe (MRS), reinforced clostridial medium (RCM) and lysogeny broth (LB) agar plates under anaerobic conditions. After incubation for 48 h, the number of colony-forming units (CFUs) was enumerated, and the density of microbes was expressed as CFU/g of feces.

### 2.6. Identification of Relevant Fecal Microbes

Based on the results of the plate count, colonies grown on MRS plates cultured with fecal samples isolated from the OVADIO group were harvested. Total DNA was extracted from the colonies using a High-Speed Plasmid Mini Kit (Geneaid Biotech Ltd., Sijhih City, Taiwan). The 16S rRNA gene was amplified by PCR with the universal primers: 16F27 (50-AGAGTTTGATCCTGGCTCAG-30) and 16R1522 (50-AAGGAGGTGATCCAGCCGCA-30) [24]. Each amplified PCR product with a size of 1.5 kb was sequenced using a 3730 DNA analyzer (Applied Biosystems, Foster City, CA, USA) and the obtained sequence was compared with microbial sequences available in the GenBank using BLAST (http://www.ncbi.nlm.nih.gov/BLAST; accessed on 20 November 2019). The results of similarity analyses of distinct strains based on 16S rRNA gene sequences are shown in Table 1. In total, five different strains were identified and kept for further cell culture experiments.

### 2.7. Cell Culture Experiments

Caco-2 cells were seeded onto 24-well plates and grown for 7 days. These strains were cultured in broth for 24 h, and the bacterial pellets were washed with sterile phosphate-buffered saline (PBS) three times for co-culture assays. A previously described method for the co-culture of Caco-2 cells with bacteria was employed [25]. Briefly, Caco-2 cells (10^6^ cells/mL) were incubated for 14 h in fresh medium containing isolated fecal microbes (10^7^ CFU/mL for each strain). The supernatants of cultured Caco-2 cells were harvested for the measurement of interleukin (IL)-8, thymic stromal lymphopoietin (TSLP), C–C motif chemokine ligand (CCL)-2, CCL-5, CCL-20 and C–X–C motif chemokine ligand (CXCL)-1 production by enzyme-linked immunosorbent assay (ELISA; Thermo Fisher Scientific Inc., Waltham, MA, USA and Biolegend, San Diego, CA, USA) following the supplier’s instructions.

The mice in the OVA group were sacrificed after six rounds of challenge, and the MLNs were collected and processed into single cell suspensions (OVA-primed MLN cells). The OVA-primed MLN cells (10^6^ cells/mL) were seeded onto 96-well culture plates treated with isolated fecal microbes (10^7^ CFU/mL for each strain) and then re-stimulated with OVA (50 μg/mL) for 24 h. Supernatants of cultured MLN cells were harvested for the measurement of IL-2, IL-4, IL-10, IL-17, interferon (IFN)-γ and transforming growth factor (TGF)-β production by ELISA (Thermo Fisher Scientific Inc. and Biolegend) following the supplier’s instructions.

A previously described in vitro method for investigating the impact of diosgenin on cells was employed [26]. Briefly, Caco-2 cells or OVA-primed MLN cells were incubated with various concentrations of diosgenin (0–200 μM) for 14 and 24 h, respectively. The supernatants were harvested for cytokine detection, as described above, and the cell viability was determined by 3-(4,5-dimethylthiazol-2-yl)-2,5-diphenyltetrazolium bromide (MTT) assay, as described previously [26].

### 2.8. Statistical Analysis

The data are expressed as the mean ± standard error of mean (SEM) for each treatment group. The three groups were compared by one-way analysis of variance (ANOVA). Two-tailed Student’s t-test was used to assess the statistical difference between the treatment group and the control group. Significance was considered as *p* < 0.05.

## 3. Results

### 3.1. Severity of Allergic Diarrhea and Density of Fecal Bacteria

An OVA-induced murine model of food allergy, which was utilized to investigate the anti-allergic effect of diosgenin and to explore the role of gut dysbiosis in food allergies, was also employed in this study to evaluate the association between diarrhea severity and fecal microbiota (Figure 1). As shown in Figure 2A, normal fecal conformation was observed in the NA group throughout the experiment. However, the mean diarrhea scores of mice in the OVA group were gradually increased, with the highest score of three at the sixth challenge, indicating the successful induction of allergic diarrhea. The mean scores of mice in the OVADIO group were lower than those of mice in the OVA group after each challenge, indicating that diosgenin treatment could ameliorate the severity of allergic diarrhea. Except for mice in the NA group, the lowest mean score of diarrhea was observed in mice in the OVADIO2 and OVADIO4 groups. Additionally, milder severity of villus edema, crypt hyperplasia and cell infiltration was observed in the duodenal tissue of the OVADIO group compared to that of the OVA group (Figure 2B), suggesting that the oral administration of diosgenin could attenuate allergic enteritis. The profile of the fecal microbiota was firstly evaluated using plating methods. As previous studies have shown different fecal bacterial counts on MRS, RCM and LB agar plates between normal and allergic mice, the fecal samples from each group were serially diluted and cultured on MRS, RCM and LB agar plates to determine the density of fecal microbes [4,27]. In line with the results of previous studies, OVA challenge significantly reduced the density of microbe growth on MRS agar plates and increased the density of microbe growth on RCM and LB agar plates (Figure 2C) [4,27]. Nevertheless, diosgenin treatment markedly reversed the diminished density of microbe growth on MRS agar plates and reduced the density of microbe growth on RCM and LB agar plates (Figure 2C). These results indicate that diosgenin treatment selectively modulated the composition of fecal microbes in allergic mice.

### 3.2. NGS Analysis of Fecal Microbes

For a comprehensive understanding of the diversity and composition of the gut microbiota, metagenomic studies were performed by analyzing the 16S rRNA gene extracted from fecal samples. As rarefaction curves are a representation of the species richness for a given number of individual samples, the species richness of the fecal samples of OVADIO2 and OVADIO4 was the highest compared to the other samples (Figure 3A), revealing a positive correlation between the species richness of fecal microbes and the diosgenin-mediated attenuation of allergic diarrhea. Community bar-plot analysis shows the abundance of sequences at the family and genus levels in the three groups (Figure 3B,C). In parallel with the results of the rarefaction curves, the highest abundance of fecal microbes was observed in the samples from the OVADIO2 and OVADIO4 groups. The abundance of *S24_7*, *Bacteroidaceae*, *Prevotellaceae*, *Lactobacillaceae*, *Bacteroides* and *Lactobacillus* was obviously elevated in the samples from the OVADIO2 and OVADIO4 groups, indicating that the anti-allergic effect of diosgenin may be associated with the growth of the above microbes. Moreover, significant differences were observed in the similarity of bacterial membership and communities in the feces of mice from these three groups (Figure 3D,E). Interestingly, the microbiota profile of OVADIO1 was not clustered with others in the same group (Figure 3D). As the diarrhea severity score of OVADIO1 was also the highest in this group (Figure 2A), it is suggested that, in addition to diosgnein treatment, individual variation of intestinal immunity may influence the composition of gut microbiota. Based on the profile of the fecal microbiota, fecal strains from diosgenin-treated mice grown on MRS plates were isolated and identified. As shown in Table 1, sequences of colonies mainly belonged to lactobacilli. Moreover, most of the isolated strains from the NA group grown on MRS plates belonged to *Lactobacillus murinus*. Taken together, these findings suggest that diosgenin treatment reshaped the gut microbiota rather than restoring the gut microbiota to that observed in normal mice.

### 3.3. Impact of Fecal MRS Strains from Diosgenin-Treated Mice on Cytokine/Chemokine Production by Caco-2 Cells

The impact of these fecal strains from diosgenin-treated mice on the intestinal cytokine/chemokine network was evaluated in Caco-2 cells. As IL-8 is a crucial chemokine to recruit immune cells from intravascular to interstitial sites, the secretion of IL-8 from Caco-2 cells co-cultured with individual strains was investigated [28]. Surprisingly, all of these strains were able to promote the secretion of IL-8 to various degrees (Figure 4A). As enterocyte-derived TSLP could promote the development of tolerogenic dendritic cells, suppress the inflammatory potential of dendritic cells in response to bacteria and maintain intestinal homeostasis, the level of TSLP in the supernatants of cultured Caco-2 cells was determined [29,30,31]. In response to the isolated strains, a markedly increased secretion of TSLP was observed (Figure 4B). Furthermore, CCL-2, CCL-5, CCL-20 and CXCL-1, secreted by enterocytes, are known to cause inflammation [32]. Therefore, the levels of these chemokines were measured. In contrast to the results of TSLP, the isolated strains limitedly induced the production of these chemokines (data not shown; lower than the detection limit).

### 3.4. Impact of Fecal MRS Strains from Diosgenin-Treated Mice on Cytokine Production by Allergen-Primed MLN Cells

As a typical food allergy involves a Th1/Th2 imbalance, the profiles of allergen-specific T-cell responses were evaluated by investigating the production of Th subset-associated cytokines from OVA-primed MLN cells re-stimulated with OVA [1,17]. As expected, higher levels of IL-2, IFN-γ (the major Th1 cytokines) and IL-4 (the major Th2 cytokine) were observed in MLN cells co-cultured with these strains than in those co-cultured without isolated strains (Figure 5A–C) [33,34]. Accordingly, the impact of isolated strains on intestinal T-cell responses did not only result from the modulation of the Th1/Th2 immune balance. As regulatory T (Treg) cells and Th17 cells have contrasting effects on intestinal inflammation, the levels of IL-10, TGF-β (the major inhibitory cytokines secreted by Treg cells) and IL-17 (the major cytokine secreted by Th17 cells) were measured [34]. Co-culture with these strains remarkably upregulated IL-10 and TGF-β production but downregulated IL-17 production by MLN cells (Figure 5D–F), indicating a shift in the T-cell response toward Treg polarization.

### 3.5. Impact of Diosgenin on Caco-2 Cells and Allergen-Primed MLN Cells

To understand whether diosgenin could modulate the cytokine/chemokine network and T-cell responses, Caco-2 cells and OVA-primed MLN cells were also incubated with various concentrations of diosgenin. Interestingly, diosgenin at concentrations greater than 5 μM reduced IL-8 and IL-2 production in a concentration-dependent manner (Figure 6A,C). As a previous study has reported the cytotoxic effect of diosgenin, the impact of diosgenin on the viability of Caco-2 cells and OVA-primed MLN cells was further explored [26]. Concordantly, diosgenin at concentrations greater than 20 μM diminished cell viability in a concentration-dependent manner (Figure 6B,D). Based on the results shown in Figure 6, it should be emphasized that diosgenin, unlike the isolated fecal strains, could not directly modulate the functionality of intestinal epithelial cells and immune cells. The in vivo anti-allergic and immunomodulatory effects of diosgenin most likely result from its beneficial influence on the gut microbiota.

## 4. Discussion

Although the regulatory effect of diosgenin on the intestinal microbiota in tumor models has been reported, the influence of diosgenin on gut dysbiosis in allergic mice and its correlation with the anti-allergic effect remains unclear [18]. In this study, we provide the first evidence to demonstrate that the anti-allergic diarrhea effect of diosgenin occurs via its improvement of gut dysbiosis. This notion has been substantiated by several lines of evidence, as shown in Scheme 1. Firstly, the density of fecal bacteria growth on the MRS plates was lower, whereas growth on RCM and LB plates was higher in mice in the OVA group compared to those in the NA group. However, diosgenin treatment reversed the density of fecal microbe growth on these agar plates. Secondly, the results of the NGS analysis show that less diversity and abundance of fecal microbes was observed in OVA-induced allergic mice. Nonetheless, treatment with diosgenin increased the diversity and abundance of fecal microbes in allergic mice, especially in those with less severe diarrhea. Moreover, a higher abundance of sequences belonging to *S24_7*, *Bacteroidaceae*, *Prevotellaceae*, *Lactobacillaceae*, *Bacteroides* and *Lactobacillus* and a lower abundance of sequences belonging to *Rikenellaceae* and *Paraprevotellaceae* were observed in these mice. Thirdly, co-culture with relevant fecal strains from diosgenin-treated mice augmented the secretion of IL-8 and TSLP but inhibited the production of CCL-2, CCL-5, CCL-20 and CXCL-1 by Caco-2 cells. Finally, these relevant fecal strains suppressed allergen-induced Th1, Th2 and Th17 immune responses but elicited Treg immunity, as evidenced by the profile of Th subset-associated cytokine production in OVA-primed MLN cells. Notably, diosgenin, unlike these fecal strains, could not modulate cytokine production and even reduced cell viability.

Variations in the gut microbiota are associated with the induction of several diseases and are strongly linked to the development of atopic disorders, including food allergies [4,35,36]. In line with the hygiene hypothesis, a reduced or altered microbial load may insufficiently counterbalance the Th2 response, thus favoring the occurrence of food allergies [37]. In our recent study employing the same murine model [4], relevant fecal microbes from allergic mice isolated from RCM and LB agar plates were defined. Most of these strains, belonging to *Enterococcus*, *Streptococcus* and *Vagococcus*, reduced the viability of Caco-2 cells but increased the production of inflammatory chemokines. Moreover, cell expansion and the secretion of IL-2, IFN-γ, IL-4 and IL-17 by MLN cells were augmented, whereas the production of IL-10 and TGF-β was diminished [4]. Herein, the same methods and models were employed to address the correlation between the diosgenin-mediated attenuation of allergic diarrhea and improvement of gut dysbiosis. In contrast to the pro-inflammatory properties of relevant microbes isolated from allergic mice, the relevant lactobacilli isolated from diosgenin-treated mice exhibited anti-inflammatory activities. Interestingly, most of the lactobacilli isolated from the NA group belonged to *Lactobacillus murinus*, as did one of the strains isolated from the OVADIO group. Compared to the other MRS strains, *L. murinus* was less able to shift the intestinal microenvironment toward anti-inflammation. As diosgenin decreased the ratio of gut microbes with pro-inflammatory and anti-inflammatory properties, the anti-diarrhea and anti-inflammatory effects of diosgenin on intestinal epithelial cells and immune cells may be attributed to the improvement of gut dysbiosis.

In parallel with the results of NGS obtained from the current study, *Rikenellaceae* species were found in OVA-sensitized mice but not in PBS-sensitized mice [3]. Conversely, several strains of *Lactobacillus* are well-known probiotics against food allergies [15,17]. Additionally, it has been suggested that *Bacteroidetes* serves as a kind of tolerance-inducing adjuvant for ingested allergens [38]. However, limited information pertaining to the association between the families of *S24-7*, *Bacteroidaceae* and *Paraprevotellaceae* and food allergy is available. In the immune-priming phase of collagen-induced arthritis, significantly reduced levels of *S24-7*, *Bacteroidaceae* and *Paraprevotellaceae* were observed, suggesting their potential to inhibit inflammation [39]. Due to the limitations of traditional culture methods, most of the relevant enteric bacteria detected by NGS could not be isolated and cultured in this study. Further efforts should be made to develop novel isolation and culture methods to illustrate the impact of the other relevant bacterial families against intestinal inflammation and allergies.

With respect to the relevant strains isolated from diosgenin-treated mice, *L. murinus* has recently been demonstrated as a potential probiotic against food allergies [16]. In a previous study, *L. animalis* LA4 was able to survive gastrointestinal passage and transitorily colonize the canine intestine, where it could positively influence the composition and metabolism of the intestinal microbiota [40]. The preventive effect of *L. salivarius* against atopic dermatitis and allergic airway responses has also been reported, and the underlying mechanism may involve modulating Th1/Th2 and Treg immunity [41,42]. However, limited information pertaining to the biological activities of *L. faecis* and *L. apodemi* is available. Although the current data obtained with individual strains in vitro are unlikely to adequately represent reactions to mixtures of gut microbiota in vivo, illustrating the impact of dominant species on intestinal epithelial cells and MLN cells may help to identify the underlying mechanisms of diosgenin-mediated anti-allergic and anti-inflammatory effects. Further investigation is required to elucidate the in vivo impact of gut dysbiosis on the epigenetic expression of T-cells and enterocytes and the in vivo effects of these isolated fecal strains against allergic enteritis, which will provide direct evidence to support the in vitro data obtained in this study.

## 5. Conclusions

In this study, we provide the first evidence demonstrating that the anti-allergic diarrhea effect of diosgenin is exerted via improving gut dysbiosis, revealing critical insights into the underlying mechanisms of diosgenin’s action against food allergies and, probably, against other immune and gastrointestinal disorders. Most importantly, these results will be essential for future studies to clarify whether the improvement of gut dysbiosis is a potential action mechanism for bioactive food or medicinal ingredients with low oral bioavailability.

## Data Availability

The data presented in this study are available on request from the corresponding author.

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
