# Peer review of "Anti-Allergic Diarrhea Effect of Diosgenin Occurs via Improving Gut Dysbiosis in a Murine Model of Food Allergy"

_molecules, 2021, doi:10.3390/molecules26092471_

Round 1

Reviewer 1 Report

The authors investigated the anti-allergic diarrhea effect of diosgenin is via Improving gut dysbiosis in a murine model of food allergy. The manuscript was well-written, and the results are sound. There are some concerns that should be addressed.

1.Some figures were big, some figures are small, some resolution of figures were low.

2.The abstract should be re-written.

3.Why the author used the murine model for allergy test?

4.Introduction should be increase for food allergy.

5.What is the concentration used in this experiment?

6.I would suggest a professional English revision once, in most of the sections throughout the manuscript, the text is almost unreadable. There are a lot of sentences that are ambiguous and should be rephrased.

7.More recent references are needed.

8.Why improving the gut dysbiosis can decrease the allergy?

Author Response

Response to the Reviewers' comments:

We have studied carefully the comments from the Reviewers and revised the manuscript by taking all issues mentioned in the Reviewers into consideration. We highlighted all the revisions in red in the revised manuscript. The revised manuscript has undergone English language editing by MDPI Specialist Edit Service. The text has been checked for correct use of grammar and common technical terms, and edited to a level suitable for reporting research in a scholarly journal. Following is a detailed point-by-point explanation on how we have addressed all the Reviewers’ concerns. The Reviewer’s comments are in black, and our responses are in blue.

Reviewer 1: The authors investigated the anti-allergic diarrhea effect of diosgenin is via Improving gut dysbiosis in a murine model of food allergy. The manuscript was well-written, and the results are sound. There are some concerns that should be addressed.

  1. Some figures were big, some figures are small, some resolution of figures were low.

The authors thank the Reviewer's suggestion. We followed the Reviewer's suggestion and modified the size and resolution of all figures in the revised manuscript.

  1. The abstract should be re-written.

The authors thank the Reviewer's suggestion. The abstract has been re-written and undergone English language editing by MDPI Specialist Edit Service (line 11-26).

  1. Why the author used the murine model for allergy test?

The authors thank the Reviewer for pointing out the important issue. In our recent study employing the same murine model of food allergy, the fecal microbiota displayed a significantly different profile between normal and allergic mice as observed by NGS analysis. The relevant fecal microbes involved in allergen-induced dysbiosis promoted the intestinal cytokine/chemokine network and allergen-specific T-cell immune responses [4]. On the other hand, the anti-allergic effect of diosgenin was also substantiated in the same murine model, although diosgenin treatment (200 mg/kg b.w. daily) only par-tially attenuated allergic diarrhea [13]. Therefore, the same murine model was em-ployed to investigate whether the anti-allergic-diarrhea effect of diosgenin is mediated by improving gut dysbiosis (line 89-97).

  1. Introduction should be increase for food allergy.

The authors concurred the Reviewer’s suggestion and provided more introduction references of food allergy in the revised manuscript (line 47-63, 89-97).

  1. What is the concentration used in this experiment?

The authors thank the Reviewer for pointing out the important issue. The dosage of diosgenin employed in the animal experiment was according to our previous study [13]. Oral administration of diosgnein (200 mg/kg b.w. daily) attenuated the allergic responses and occurrence of allergic diarrhea in the same murine model of food allergy. However, not all allergic mice treated with diosgenin could be protected from allergic diarrhea (line 94-95, 132, 260). The concentration of bacteria (107 CFU/mL), Caco-2 cells (106 cells/mL) and MLN cells (106 cells/mL) employed in the cell culture experiments was according to the studies of Huang et al. and Chiu et al. [4, 25] (line 202-203, 211-212). Due to the cytotoxic effect of diosgenin [26], we investigated the impact of diosgenin at various concentrations (0-200 μM) on Caco-2 cells and MLN cells (line 217-219). Diosgenin at the concentrations greater than 20 μM diminished cell viability, which is consistent with the results of previous study [26] (line 369-371).   

  1. I would suggest a professional English revision once, in most of the sections throughout the manuscript, the text is almost unreadable. There are a lot of sentences that are ambiguous and should be rephrased.

The authors thank the Reviewer's suggestion. The revised manuscript has undergone English language editing by MDPI Specialist Edit Service. The text has been checked for correct use of grammar and common technical terms, and edited to a level suitable for reporting research in a scholarly journal.

  1. More recent references are needed.

The authors concurred the Reviewer’s suggestion. There are 8 recent references added in the revised manuscript (line 487-504).

  1. Why improving the gut dysbiosis can decrease the allergy?

The authors thank the Reviewer for pointing out the important issue. In our recent study employing the same murine model of food allergy, the fecal microbiota displayed a significantly different profile between normal and allergic mice as observed by NGS analysis. The relevant fecal microbes involved in allergen-induced dysbiosis promoted the intestinal cytokine/chemokine network and allergen-specific T-cell immune responses [4] (line 89-93). In addition, several studies have reported the beneficial effects of probiotics on improving allergy and gut microbiota [3, 4, 35, 36]. As the results of current study show that the fecal MRS strains suppressed inflammatory chemokine and allergen-specific T-cell cytokine secretion but increased Treg-cell cytokine production, we suggest that improvement of gut dysbiosis may shift the intestinal microenvironment toward anti-inflammation and thus attenuate allergic enteritis (line 342-354). On the other hand, many studies have found that environmental factor-induced allergic or anti-allergic effects are mediated by epigenetic mechanisms [5–9] (48-50). Therefore, further investigation is required to elucidate the in vivo impact of gut dysbiosis on the epigenetic expression of T- cells and enterocytes and the in vivo effects of these isolated fecal strains against allergic enteritis, which will provide direct evidence to support the in vitro data obtained from in this study. (462-465)

Reviewer 2 Report

This paper (on an interesting subject) suffers from weakness at different points.

The number of animals in each study group is not clearly stated. Means 8 animals, 8 in each group or the total number?

RCM medium is not a selective medium for anaerobes, enterobacteria will grow on this medium but also on LB medium which makes it difficult to distinguish anaerobes from enterobacteria.Lactobacilli can also grow.

Isolating lactobacilli already known as probiotics is not surprising, it would have been interesting to define and to analyze also the inflammatory part of the microbiome implicated in the used model of food allergy.

Pooling samples for bacteriological analysis seems not sound, it is better to analyze individual samples. Animals even in animal facilities can have individual reactions.

Can lactobacilli isola ted from the NA group give the same results as lactobacilli from diosgenin treated mice?

Bacterial counts about 5 log CFU/g (figure 2) is rather surprising as in feces always sufficient substrates are available for abundant growth and lactobacilli will also grow up.

The use of animal models  should be discussed and modulate the obtained results.

What do you want to do ? New mailCopy

Author Response

Response to the Reviewers' comments:

We have studied carefully the comments from the Reviewers and revised the manuscript by taking all issues mentioned in the Reviewers into consideration. We highlighted all the revisions in red in the revised manuscript. The revised manuscript has undergone English language editing by MDPI Specialist Edit Service. The text has been checked for correct use of grammar and common technical terms, and edited to a level suitable for reporting research in a scholarly journal. Following is a detailed point-by-point explanation on how we have addressed all the Reviewers’ concerns. The Reviewer’s comments are in black, and our responses are in blue.

Reviewer 2: his paper (on an interesting subject) suffers from weakness at different points.

  1. The number of animals in each study group is not clearly stated. Means 8 animals, 8 in each group or the total number?

The authors thank the Reviewer for pointing out the important issue. Eight mice per group was employed in the animal experiment (line 134).

  1. RCM medium is not a selective medium for anaerobes, enterobacteria will grow on this medium but also on LB medium which makes it difficult to distinguish anaerobes from enterobacteria. Lactobacilli can also grow. Isolating lactobacilli already known as probiotics is not surprising, it would have been interesting to define and to analyze also the inflammatory part of the microbiome implicated in the used model of food allergy.

The authors agree with the Reviewer's comment. In our recent study employing the same murine model [4], relevant fecal microbes from allergic mice isolated from RCM and LB agar plates were defined. Most of these strains, belonging to Enterococcus, Streptococcus and Vagococcus, reduced the viability of Caco-2 cells but increased the production of inflammatory chemokines. Moreover, cell expansion and the secretion of IL-2, IFN-γ, IL-4 and IL-17 by MLN cells were augmented, whereas the production of IL-10 and TGF-β was diminished [4]. Herein, the same methods and models were employed to address the correlation between the diosgenin-mediated attenuation of allergic diarrhea and improvement of gut dysbiosis. In contrast to the pro-inflammatory properties of relevant microbes isolated from allergic mice, the relevant lactobacilli isolated from diosgenin-treated mice exhibited anti-inflammatory activities. Interestingly, most of the lactobacilli isolated from the NA group belonged to Lactobacillus murinus, as did one of the strains isolated from the OVA-DIO group. Compared to the other MRS strains, L. murinus was less able to shift the intestinal microenvironment toward anti-inflammation. Because diosgenin decreased the ratio of gut microbes with pro-inflammatory and anti-inflammatory properties, the anti-diarrhea and anti-inflammatory effects of diosgenin on intestinal epithelial cells and immune cells may be attributed to the improvement of gut dysbiosis (line 413-429).

  1. Pooling samples for bacteriological analysis seems not sound, it is better to analyze individual samples. Animals even in animal facilities can have individual reactions.

The authors thank the Reviewer for pointing out the important issue. Because of the expensive costs of NGS service and insufficient funding, the fecal samples from two mice with similar diarrhea severity scores were pooled to reduce the number of samples for NGS analysis. The results of both the current and previous studies, employed the same models and methods, showed significantly different profile of fecal microbiota by NGS analysis of pooled samples [4].

  1. Can lactobacilli isolated from the NA group give the same results as lactobacilli from diosgenin treated mice?

The authors agree with the Reviewer's comment. Most of the lactobacilli isolated from the NA group belonged to Lactobacillus murinus, as did one of the strains isolated from the OVA-DIO group. Compared to the other MRS strains, L. murinus was less able to shift the intestinal microenvironment toward anti-inflammation. Therefore, we suggest the ratio of gut microbes with pro-inflammatory and anti-inflammatory properties has a great impact on the severity of allergic enteritis and allergic diarrhea (line 423-426).

  1. Bacterial counts about 5 log CFU/g (figure 2) is rather surprising as in feces always sufficient substrates are available for abundant growth and lactobacilli will also grow up.

The authors thank the Reviewer's suggestion. In the previous studies employing the same murine model of food allergy and the same bacterial culture method, the range of bacterial counts were from 5 log CFU/g to 8 5 log CFU/g [4, 8-10]. In parallel, the bacterial counts observed in this study were higher than 5 log CFU/g. The bacterial counts of lactobacilli of OVADIO group were even higher than 8 log CFU/g.

  1. The use of animal models should be discussed and modulate the obtained results.

The authors concurred the Reviewer’s suggestion. The discussions were added in the revised manuscript, and the description of results were revised (line 413-429).

Reviewer 3 Report

With interest, I read the manuscript molecules-1108571. Overall, it is an interesting work. I have several comments though.

Comments:

  1. All names of the experimental groups of animals should be explained in the main text as well or in Figure 1 and 2 and it should be clearly stated in the main text that the abbreviations are explained there. At the moment, there is complete confusion, e.g. “OVA/DIO”, “OVA-DIO”, and “OVADIO” most probably referring to the same group are used in the different paces in this work.
  2. Furthermore, abbreviations such as “OVADIO1”, “OVADIO2”, etc. appear. Do the correspond to different animals within a group? The same for “OVA1”, “OVA2”, “NA2”, “NA2”, etc.?
  3. You state you used 8 animals, both in the main text and the legend to Figure 1. But if you summarize, it looks you used 12, i.e. 3 groups 4 animals each = 12 animals. Right?
  4. Why only four animals per group were used, not e.g. 6 or 8?
  5. “The data are expressed as the mean ± standard error of mean (SEM) for each treatment group. The three groups were compared by one-way analysis of variance (ANOVA). Dunnett’s two-tailed t-test was used to assess the statistical difference between the treatment group and the control group.”. Dunnett’s test is a post-hoc test following ANOVA used when you have 3 groups or more. However, t-test should be used for pairwise comparisons. Zou need to rewrite these two sentences so that they make sense.
  6. Figure 3D. Why one animal (“OVADIO1”) does not cluster?
  7. Figure 4. What about other plates you used?
  8. The contribution of the bacteria seems to be crucial to your hypothesis summarized in Scheme 1, right? If so, and the allergen exposure and especially diosgenin exert their effects through bacteria, the experiments the results of which are reported in Figure 6 do not make sense, right?
  9. You refer to the role of T-cell responses in food allergy. Please, refer to PMID: 33086571 to shortly discuss the role of epigenetic changes in T cells in the development of food allergy.
  10. In addition, please, refer to PMID: 31349704 to discuss the role of T cells and MLNs in mediating the development of protection against food allergy development.

Other comments:

  1. All abbreviations need to be explained upon their first appearance, e.g. “MLN” in the Abstract.
  2. Figure 5. It should be more clearly visible which graphs refer to Caco-2 cells and which to MLNs.

Author Response

Response to the Reviewers' comments:

We have studied carefully the comments from the Reviewers and revised the manuscript by taking all issues mentioned in the Reviewers into consideration. We highlighted all the revisions in red in the revised manuscript. The revised manuscript has undergone English language editing by MDPI Specialist Edit Service. The text has been checked for correct use of grammar and common technical terms, and edited to a level suitable for reporting research in a scholarly journal. Following is a detailed point-by-point explanation on how we have addressed all the Reviewers’ concerns. The Reviewer’s comments are in black, and our responses are in blue.

Reviewer 3

With interest, I read the manuscript molecules-1108571. Overall, it is an interesting work. I have several comments though.

Comments:

  1. All names of the experimental groups of animals should be explained in the main text as well or in Figure 1 and 2 and it should be clearly stated in the main text that the abbreviations are explained there. At the moment, there is complete confusion, e.g. “OVA/DIO”, “OVA-DIO”, and “OVADIO” most probably referring to the same group are used in the different paces in this work.

The authors thank the Reviewer for pointing out the important issue. All names of the experimental groups of animals and the abbreviations were corrected and explained in the revised manuscript. 

  1. Furthermore, abbreviations such as “OVADIO1”, “OVADIO2”, etc. appear. Do the correspond to different animals within a group? The same for “OVA1”, “OVA2”, “NA2”, “NA2”, etc.?

The authors thank the Reviewer for pointing out the important issue. Because of the expensive costs of NGS service and insufficient funding, the fecal samples from two mice with similar diarrhea severity scores were pooled to reduce the number of samples for NGS analysis. Therefore, there were 4 fecal samples per group, and the samples were labeled as 1, 2, 3 and 4 (i.e., the fecal samples of the OVADIO group were named OVADIO1, OVADIO2, OVADIO3 and OVADIO4) (line 147-149).

  1. You state you used 8 animals, both in the main text and the legend to Figure 1. But if you summarize, it looks you used 12, i.e. 3 groups 4 animals each = 12 animals. Right? Why only four animals per group were used, not e.g. 6 or 8?

The authors thank the Reviewer for pointing out the important issue. Eight mice per group was employed in the animal experiment (line 129-130). Because of the expensive costs of NGS service and insufficient funding, the fecal samples from two mice with similar diarrhea severity scores were pooled to reduce the number of samples for NGS analysis. Therefore, there are total 12 fecal samples for analysis.

  1. The data are expressed as the mean ± standard error of mean (SEM) for each treatment group. The three groups were compared by one-way analysis of variance (ANOVA). Dunnett’s two-tailed t-test was used to assess the statistical difference between the treatment group and the control group.”. Dunnett’s test is a post-hoc test following ANOVA used when you have 3 groups or more. However, t-test should be used for pairwise comparisons. Zou need to rewrite these two sentences so that they make sense.

The authors thank the Reviewer for pointing out the important issue. The description of Statistical Analysis was corrected in the revised manuscript (line 226).

  1. Figure 3D. Why one animal (“OVADIO1”) does not cluster?

The authors thank the Reviewer for pointing out the important issue. The diarrhea severity score of OVADIO1 was the highest in this group, revealing vigorous allergic enteritis occurred in the mice of OVADIO1 compared to the others in the same group. In parallel, microbiota profile of OVADIO1 was not clustered with others in the same group. These results are in consistent with previous studies showing the different profile of gut microbiota between normal mice and that with food allergy [4]. Although all mice in OVADIO group were treated with diosgenin, not all the mice were protected from food allergy. Compared to the treatment of diosgnein, modulation of gut microbiota is more closely associated with the attenuation of food allergy. In addition to diosgnein treatment, other factors, such as individual variation of intestinal immunity and gene expression, may influence the composition of gut microbiota (line 293-297).

  1. Figure 4. What about other plates you used?

The authors thank the Reviewer for pointing out the important issue. In our recent study employing the same murine model [4], relevant fecal microbes from allergic mice isolated from RCM and LB agar plates were increased. Most of these strains, belonging to Enterococcus, Streptococcus and Vagococcus, reduced the viability of Caco-2 cells but increased the production of inflammatory chemokines. Moreover, cell expansion and the secretion of IL-2, IFN-γ, IL-4 and IL-17 by MLN cells were augmented, whereas the production of IL-10 and TGF-β was diminished [4]. (line 413-418).

  1. The contribution of the bacteria seems to be crucial to your hypothesis summarized in Scheme 1, right? If so, and the allergen exposure and especially diosgenin exert their effects through bacteria, the experiments the results of which are reported in Figure 6 do not make sense, right?

We thank the Reviewer for this comment. Based on the results shown in Figure 6, it should be emphasized that diosgenin, unlike the isolated fecal strains, could not directly modulate the functionality of intestinal epithelial cells and immune cells. The in vivo anti-allergic and immunomodulatory effects of diosgenin most likely result from its beneficial influence on the gut microbiota (line 371-375). Therefore, the results of Figure 6 are important to support our hypothesis.

  1. You refer to the role of T-cell responses in food allergy. Please, refer to PMID: 33086571 to shortly discuss the role of epigenetic changes in T cells in the development of food allergy. In addition, please, refer to PMID: 31349704 to discuss the role of T cells and MLNs in mediating the development of protection against food allergy development.

We thank the Reviewer for this comment to improve this manuscript. There are 8 recent references (including PMID: 33086571 and PMID: 31349704) added in the revised manuscript. Description pertaining to the role of epigenetic changes in T cells in the development of food allergy and the role of T cells and MLNs in mediating the development of protection against food allergy development was added into the revised manuscript (line 47-63).

Other comments:

All abbreviations need to be explained upon their first appearance, e.g. “MLN” in the Abstract.

We thank the Reviewer for this comment. All abbreviations are explained upon their first appearance in the revised manuscript.

Figure 5. It should be more clearly visible which graphs refer to Caco-2 cells and which to MLNs.

We thank the Reviewer for this comment. All the data shown in Figure 5 was obtained from the experiments of MLN cells, which is noted in the figure legend.

Round 2

Reviewer 3 Report

Thank you very much for addressing my comments well. I have not further reservations.